# Evaluation of Atmospheric Features in Natural Disasters due Frontal Systems over Southern Brazil

Jean Souza dos Reis [1,2,*], Weber Andrade Gonçalves [1], Diego Oliveira de Souza [3] and David Mendes [1]

1   Climate Sciences Post-Graduate Program, Department of Climate and Atmospheric Sciences, Federal University of Rio Grande do Norte, Av. Senador Salgado Filho 3000, Lagoa Nova, Natal 59078-970, Brazil
2   SENAI Institute of Innovation—Renewable Energies, Av. Capitão Mor-Gouveia 2770, Lagoa Nova, Natal 59063-400, Brazil
3   National Center for Monitoring and Early Warning of Natural Disasters, Estrada Doutor Altino Bondesan, 500, Distrito de Eugênio de Melo, São José dos Campos, Sao Paulo 12247-016, Brazil
*   Correspondence: jean.reis@ufrn.br

**Abstract:** This study evaluated the atmospheric pattern precursors to the occurrence of natural disasters (ND) in the southern region of Brazil (SRB) due to the passage of frontal systems (FS). The results can be used as prognostics to assist in risk management with a set of preventive and mitigating actions in order to minimize the impact of natural disasters suffered by the population. The natural disasters data were provided by the Centro de monitoramento e alertas de desastres naturais (Cemaden). For atmospheric analysis we used ERA5 reanalysis data, and the precipitation dataset was estimated from Integrated Multi-satellitE Retrievals for GPM (IMERG) from the Global Precipitation Measurement (GPM) mission. The most affected regions are the coast of Santa Catarina and the central-eastern region of Rio Grande do Sul. The results indicate that FS associated with ND are different from the other FS that affect the SRB. The observations were: a pattern of increase and accumulation of available convective potential energy west of the SRB before the event, especially in spring; a considerable increase in specific humidity at low levels associated with runoff east of the Andes; and an anticyclonic circulation at high levels similar to the Bolivian High. Analysis of rainfall behavior indicates that it is highest in the two days preceding the disaster. The mean precipitation values identified, together with atmospheric behavior observed in this study, allow us to identify the potential occurrence of a disaster in the cities of SRB in the passage of a frontal system.

**Keywords:** precipitation; risk; atmospheric dynamics; environmental health; climate impact

## 1. Introduction

The importance of knowing the atmospheric and climatic conditions associated with natural disasters (ND) is essential to develop new tools and products that assist with weather forecasting in the short and medium term, as well as mitigation policies for maximum possible risk reduction. In recent decades, this topic has attracted the attention of researchers, governments, decision makers and society in general due to its strong social and economic impact.

An ND is the result of a combination of four important factors, which are: (1) the occurrence of a natural hazard; (2) an exposed population; (3) the social and environmental vulnerability conditions of this population; and (4) insufficient capacities or measures to reduce the potential risks and damage to the health of the population [1–3]. The economic costs related to the impacts of ND have increased worldwide over time. Average annual values over a 10-year period increased from $4 billion per year in the 1950s, to $13 billion per year in the 1970s and to $65 billion per year in the 1990s [4,5]. These costs are mainly due to climate-related risks [6]. Between 1997 and 2017, there was a total of $2908 billion in economic impacts around the world, and around 1.3 million deaths according to the United Nations Office for Disaster Risk Reduction (UNISDR) report [7] with events such as floods

and droughts accounting for 91% of all disasters. According to Hong et al. [8], in the United States between 1980 and 200 events such as intense storms, floods and hurricanes caused more than $254 million worth of damage and 1331 lives were lost. Since 2000, the figures for these events have more than doubled to exceed more than $1 trillion [9,10]. Recently, it has been documented that the risk and the occurrence of ND has increased significantly over the past six decades [11].

Marengo [12] indicates that severe weather events will be one of the major problems facing humans in the 21st century. According to data from the US Office of Foreign Disaster Assistance (OFDA) and the Centre for Research on the Epidemiology of Disaster (CRED), most of these disasters were caused by severe atmospheric instabilities, i.e., intense atmospheric events that can cause great socio-economic damage due to episodes of intense precipitation, windstorms, hail and tornadoes [13,14].

Extreme events associated with atmospheric processes are common in the southern region of Brazil (SRB). According to the National Atlas of Natural Disasters in Brazil [15,16], extreme events related to rainfall prevail over other types of disasters. Historically, the SRB is affected not only by the occurrence of major disasters, but also by the frequency and variety of adverse events, and even by the occurrence of atypical phenomena, such as the case of Hurricane Catarina that generated losses of around US$ 1 billion in 2004 [17,18]. Numerous fatalities due to flooding and landslides have been documented. For example, Vale do Itajaí in the state of Santa Catarina (SC) in 2008, where 120 deaths were reported [19], landslides and extremes of rainfall were reported in the Itajaí river basin in 2011 and 2013 [20], and the landslides that were reported in the Serra da Prata in the state of Paraná (PR) [21]. Recent work, such as Dunn et al. [22], shows that heavy rainfall has increased in the SRB. Debortoli et al. [23], assessing disaster risks using IPCC AR5 models, found that the SRB is the region with the most significant increase (around 50%) in landslide risk.

One of the main meteorological systems, among several that contribute to the distribution of precipitation in the SRB Region and associated with adverse events, are frontal systems (FS) [24–26]. According to Catto et al. [27], FS are one of the main rainfall-causing systems in the SRB Region and they are widely studied because their effect on weather conditions is more noticeable, more frequent and easily identifiable [28]. There are several studies on the influence of FS in South America (SA) [24–26,29–33]. Due to the characteristics of atmosphere and the geographic position of SA (between the subtropical highs of the South Pacific and South Atlantic), a propitious environment is created for these systems [34]. FS reach SA throughout the year, being recorded in greater numbers at higher latitudes and they present a typical southwesterly to northeasterly displacement. The highest frequency of FS occurs between the months of May and September and the lowest frequency in summer (December to February) [26]. In the SRB, they are more frequent from June to September [35].

Previous studies have been carried out to understand the importance of meteorological phenomena which precede ND in the SRB, such as Nedel et al. [36] who zoned the ND that occurred in Rio Grande do Sul (RS) due to hail and windstorms. Other studies, such as Escobar et al. [37] who classified cold fronts associated with extreme rainfall in eastern SC, and Seluchi et al. [33] who analyzed the main characteristics of cold fronts causing intense rainfall along the coast of SC, have also contributed to the knowledge about the ND. However, the literature still needs studies that assess the influence and relationship between FS and ND across the SRB.

Considering the relevance of the topic in the region, previous studies and the necessity to improve the literature, the main objective of this study is to assess the atmospheric patterns associated with ND due to passage of FS in order to assist forecasting, monitoring and risk management centers in making decisions that can prevent events, mitigate risks and lessen the adverse impacts arising from NDs, such as material and human losses across the SRB.

## 2. Materials and Methods

### 2.1. Overview of Study Area

The SRB (Figure 1) is situated between latitudes 22° South and 34° South in a largely subtropical climate, except in the extreme north of PR, which is located in the high-altitude tropical climate. The region has well-defined seasons, with rainfall distributed throughout the year and the widest temperature range of any Brazilian region [25]. The SRB occupies an area of 564 thousand km$^2$ which corresponds to 6.76% of the Brazilian territory. It is the smallest region in the country, composed of three states, PR, SC and RS. There are 1191 municipalities and it borders three countries: Uruguay, Argentina and Paraguay. According to the Instituto Brasileiro de Geografia e Estatística (IBGE) [38], there are 27,384,815 inhabitants, making it the second most populous region in Brazil with a demographic density of 48.58 inhabitants per km$^2$. In 2010, the approximate population in risk areas in these municipalities reached 8,270,127 inhabitants and 2,471,349 households [39].

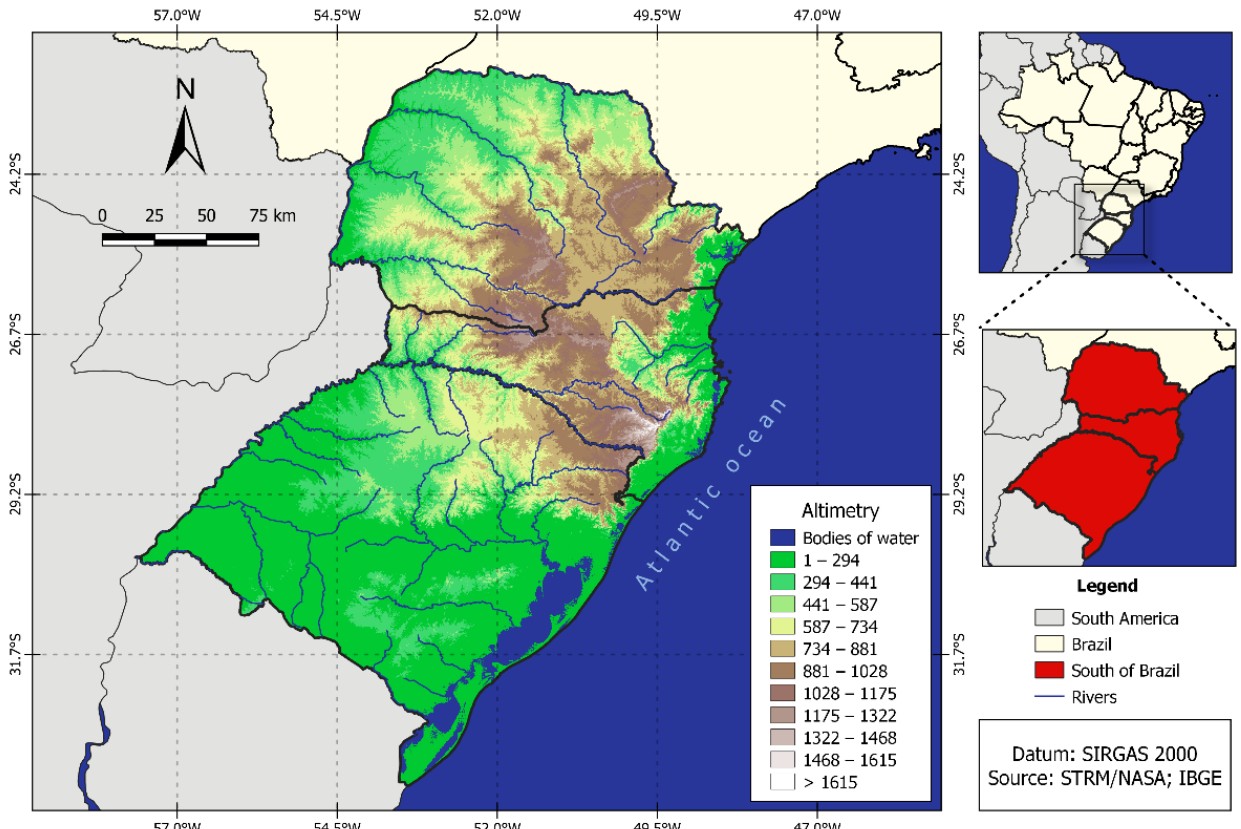

**Figure 1.** Location of the Southern Region of Brazil.

According to Grimm [25], the latitudinal and relief variability, the maritimity/continentality and the action of several tropical and extratropical systems of medium latitudes contribute to the occurrence of great contrasts in temperature and precipitation regimes. Besides the influence of the different solar radiation reaching each latitude and the important aspects of relief, the climate of the SRB is basically determined by the position and intensity of the South Atlantic subtropical high, a semi-permanent pressure system and the associated anticyclonic circulation [25].

### 2.2. Natural Disasters Data

The ND data were provided by the Centro Nacional de Monitoramento e Alertas de Desastres Naturais (Cemaden). The data is available for the cities under Cemaden monitoring, highlighted in Figure 2; more details can be found at http://www.cemaden.gov.br/municipios-monitorados-2/ (accessed on 29 July 2020). The municipalities monitored by Cemaden have a history of ND records resulting from mass movements (landslides,

mass rushes, undermining of banks/fallen lands, falling/rolling of rocky blocks and erosive processes) and/or resulting from hydrological processes (floods, mudslides, major flooding). In this study, we use ND records due to hydrological processes because of the nature of the system analyzed. The data is available from the period of 2016 to 2020 and contains information such as the type of event, date, latitude and longitude in which the disaster occurred [40,41].

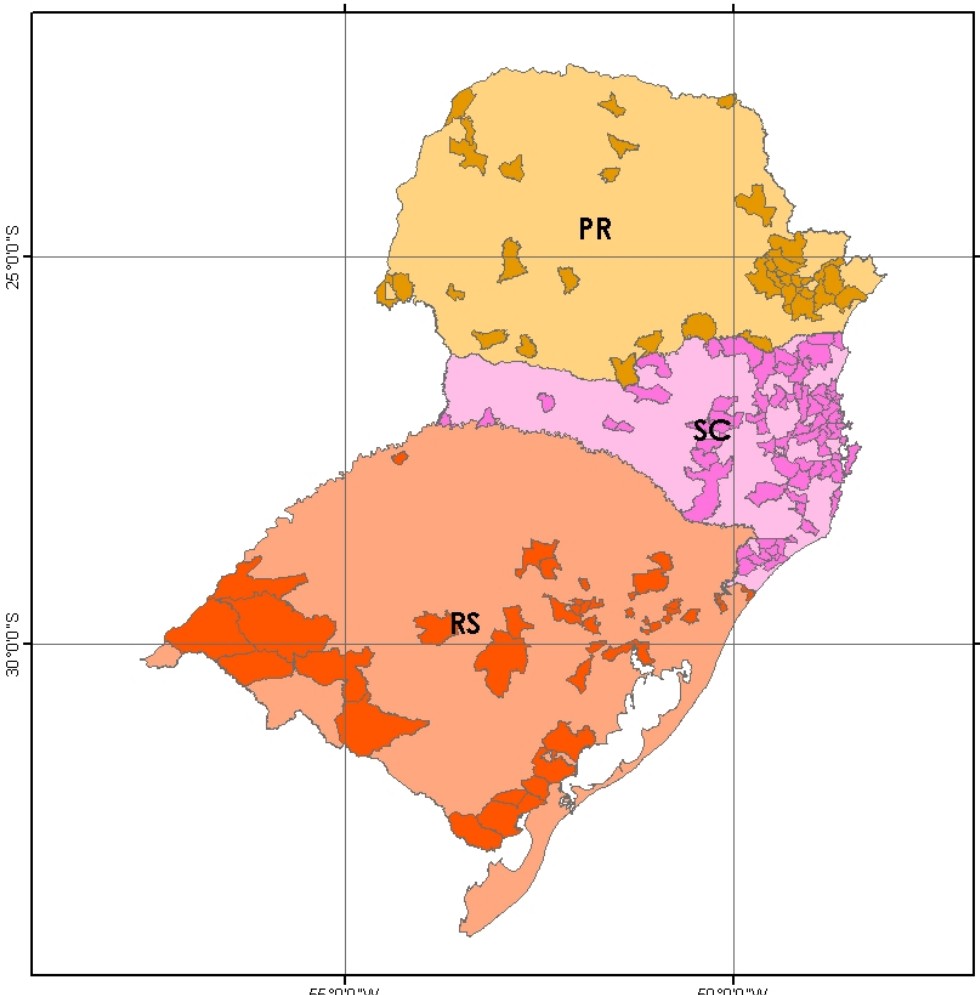

**Figure 2.** Municipalities monitored in southern region of Brazil. Source: Cemaden.

### 2.3. Meteorological Data

ERA5 is the latest reanalyzed European Centre for Medium-Range Weather Forecasts (ECMWF) (fifth generation) climate dataset [42]. Compared to its predecessor, the ERA-Interim [43], there are substantial improvements such as finer spatial grid (31 km), higher temporal resolution (3 in 3 h), increase in the number of levels (137 levels) and primarily changes in the calculation of atmospheric parameters due to the change in the assimilation system from IFS Cycle31r2 to IFS Cycle41r2. Since 2020, ERA5 data covers the period from 1950 to the present with daily 5-day lag updates from real time. In this study, the surface variables used are mean sea-level pressure (MSLP) (hPa) and Convective Available Potential Energy (J kg$^{-1}$). The variables in pressure levels are the zonal and meridional wind components (m s$^{-1}$), temperature (°C), geopotential height (m$^2$ s$^{-2}$), specific humidity (kg kg$^{-1}$) and divergence (s$^{-1}$) between the 250 and 1000 hPa with a spatial resolution of 0.25° × 0.25° and 6-h time resolution, for the period 2016 to 2020.

### 2.4. Precipitation Data

For this study, we used the daily accumulated precipitation data, estimated from the Integrated Multi-satellitE Retrievals for GPM (IMERG) (version 5) algorithm of the Global Precipitation Measurement (GPM) mission. The GPM project is the result of efforts between the United States National Aeronautics and Space Administration (NASA) and the Japan Aerospace Exploration Agency (JAXA), which began in 2014 with the intention of providing global precipitation measurements. Generally, GPM offers three different levels of data processing and level 3 products are most commonly used for scientific and operational purposes. There are three different IMERG Daily products, which include IMERG Day 1 Early Run (near real-time with 6 h latency), IMERG Day 1 Late Run (reprocessed near real-time with 18 h latency) and IMERG Day 1 Final Run products (adjusted with four-month latency) [44]. In this study, we use the IMERG Final Run level 3 product precipitation estimate, with a spatial resolution of $0.25° \times 0.25°$ and a daily temporal in the period from 2016 to 2020.

### 2.5. Analysis Methods

The atmospheric analyses were carried out through compositions and maps of meteorological variables for the days before, during and after the occurrence of ND in the monitored municipalities. According to Reboita [45], in synoptic analyses there are key variables used to assess the characteristics of atmospheric systems which are PMSL, layer thickness (which is the difference in geopotential height between two surfaces of constant pressure, and which is proportional to the average air temperature in the layer), wind speed and direction. In addition to these variables, air temperature, specific humidity, divergence, precipitation and CAPE were used to characterize the atmosphere. A case composition of ND events was created from 2 days before (day $-2$) to the day of the event (day 0), following a methodology similar to that employed by Garreaud [46], Mendes et al. [47], Cavalcanti and Kousky [35] and Seluchi et al. [33].

To confirm the presence of a FS over the SRB on the same day that a ND occurred, surface synoptic charts from the Centro de Previsão do Tempo de Estudos Climáticos/Instituto Nacional de Pesquisas Espaciais (CPTEC/INPE) were analyzed for the four main synoptic times (00, 06, 12 and 18 UTC). When FS symbology (cold, warm, occluded or stationary front) was identified over the municipality or in an adjacent region very close to where the ND occurred, it was considered that the disaster was influenced by the FS. An example can be seen in Figure S1 (Supplementary Materials).

Anomaly fields were calculated by the difference between the average composite value of the variable and the climatology of the same variable on days that the FS reached the SRB. The FS climatology was performed in a similar way to the ND analysis, identifying the presence of FS over the SRB in all the synoptic charts in the period from 2016 to 2020.

Significance Test

To assess the statistical significance of the mean compositions, Student's one-sample t-test was performed, given by:

$$t = \frac{\overline{x} - \mu_0}{s/\sqrt{n}} \tag{1}$$

where $\overline{x}$ and $s$ are the mean and variance of the composite of a given weather variable, $\mu_0$ is the climatological average of that variable and $n$ is the number of observations. Student's t-test is commonly used in climatological studies [48]. Then, the means of the compositions and climatology are compared and tested to see if they are statistically different at 99% confidence level.

## 3. Results and Discussion

### 3.1. Spatio-Temporal Analysis of Natural Disasters

This topic discusses the spatio-temporal behavior of NDs as a result of the passage of an FS over the SRB. As previously mentioned, in this study the ND that have been

analyzed are of hydrological nature, due to the characteristic atmospheric influences of the FS. There were 120 cases that occurred with the presence of a FS over the SRB on 60 separate days, equivalent to 16% of the total ND recorded by Cemaden. Figure 3 shows the spatial distribution of municipalities that were affected by ND in the period from 2016 to 2020. The most affected regions are the coast of SC and the Centre-East region of RS. Around 64.1% of the ND recorded by Cemaden were of hydrological nature (among flooding, mudslides and floods), which corroborates the National Atlas of Natural Disasters of Brazil [15,16] which states that in the SRB, between 1991 and 2012, extreme events related to the rainfall regime prevailed over other types of disasters. According to Cemaden estimates, more than 20,000 people were displaced due to the impact of the 120 NDs analyzed. An important caveat is made because the monitoring carried out by Cemaden does not cover all the municipalities of the three states belonging to the SRB, which can lead to the underreporting of ND due to the FS in other cities in the region.

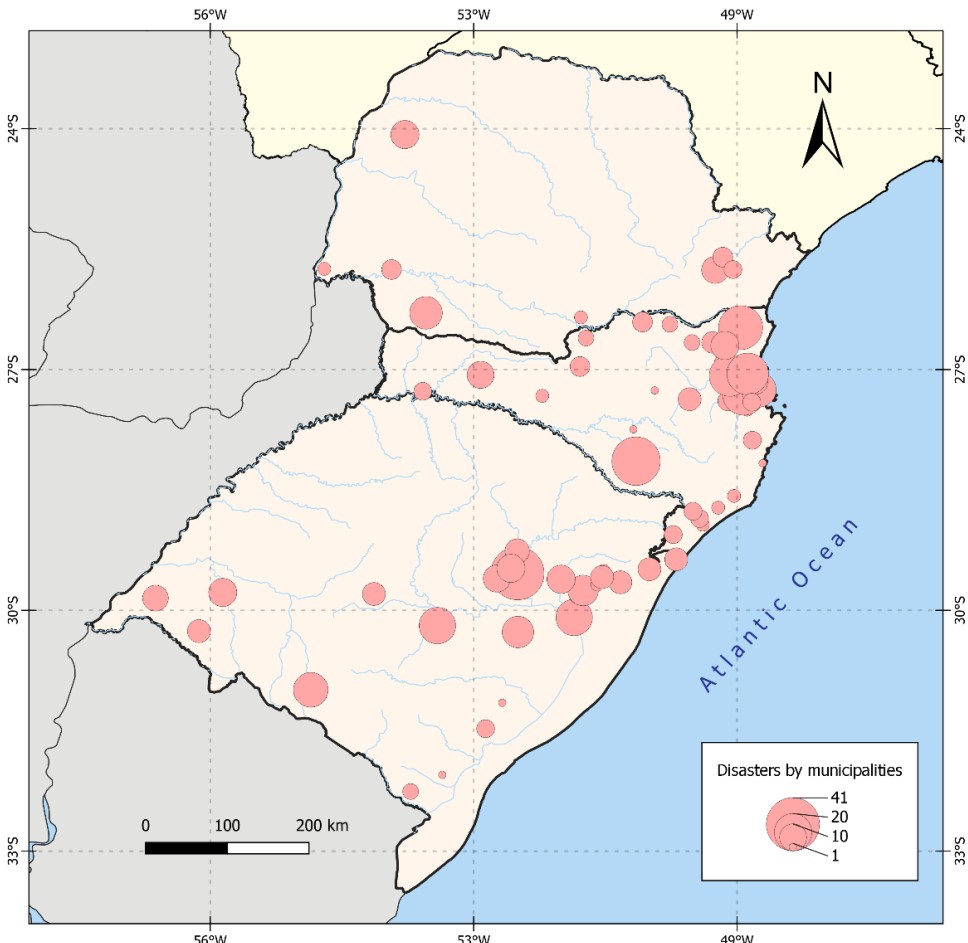

**Figure 3.** Population exposed in risk areas by municipality in the SRB.

Figure 4 shows the monthly (Figure 4a) and annual (Figure 4b) distribution of ND over the period from 2016 to 2020. Most ND occurred in May (23%), while the month of lowest occurrence was December (0.83%). There was a second peak in June (18.3%) and a third in October (15.0%). As previously mentioned, the FS and Mesoscale Convective Systems (MCS) occur throughout the year; however, there is greater activity of FS between May and September [35,49], which supports the results presented in this study. Cavalcanti [35] shows that the average number of FS passing through the SRB region is highest between July and October.

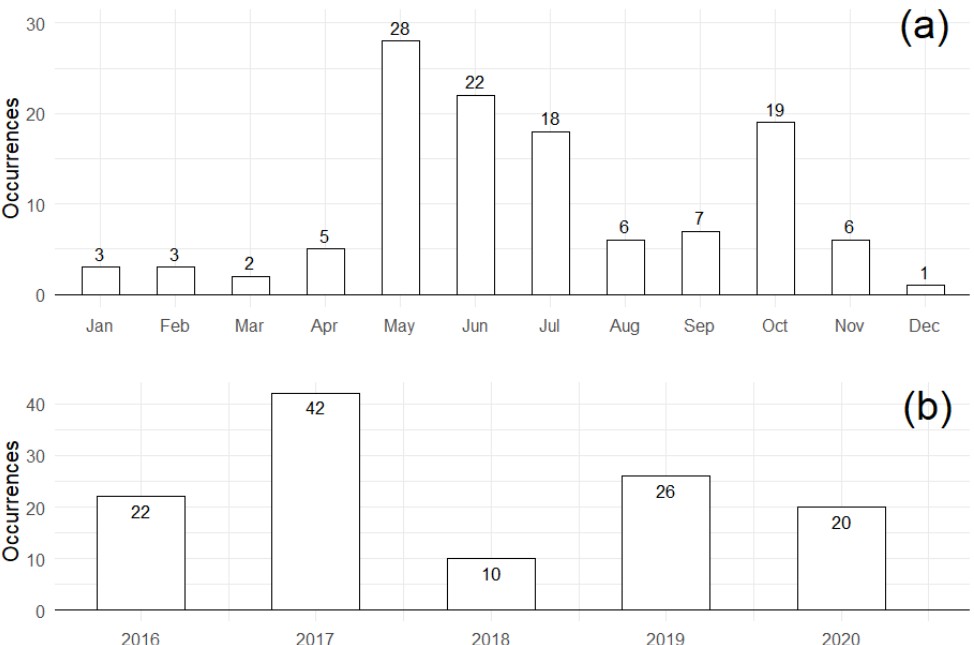

**Figure 4.** Monthly (**a**) and annual (**b**) distribution of ND in the SRB.

The annual distribution (Figure 4b) shows that most NDs occurred in the years 2017 and 2019. These years were affected by heavy rains followed by ND which were highlighted in the national media. In March 2017, storms hit the northeast region of RS, especially in the city of São Francisco de Paula, leaving 1600 people homeless and one dead [50]. Furthermore, according to the Civil Defense, there was damage to more than 500 homes in five different neighborhoods. In May 2017, heavy rainfall in the north, northwest and west regions of RS forced 10 municipalities to declare a state of emergency, while Civil Defense counted more than 56 cities with severe damage [51]. In the same month, the western region of RS had 300 mm above average rainfall [52]. In June 2017, 24 municipalities in PR, SC and RS declared a situation of emergency due to the heavy rains that occurred over the region [53,54]. In August after heavy rains, 173 cities triggered a situation of emergency in RS, and the municipality of Jaguaruna-SC decreed a state of public calamity according to data from the Ministry of National Integration [55]. October registered constant rainfall for three days that left 25 RS cities in emergency situations [56]. The rain also damaged agriculture in RS; the planting of rice, for example, was delayed. In 2016, at the same time, more than half the area was already cultivated. The machines could not enter the flooded crops due to the constant rains [56]. In 2019, the National Civil Defense recognized emergency situations in 27 cities because of various disasters [57]. In RS, three municipalities were affected by heavy rains: Canguçu, Pedras Altas and São Lourenço do Sul. In another three, Cachoeira do Sul, Dom Pedrito and Alegre, windstorms were recorded. In Itaiópolis (SC) and Viçosa (MG), there were hailstorms [57].

*3.2. Atmospheric Evaluation*

The average of the composite fields of ND due to FS by season are shown in the following figures (Figures 5–11) with the panel being presented with the following scheme: 2 days before ND (−2), 1 day before ND (−1) and ND day (0). As there are few cases in the summer season (Figure 5a), only the autumn, winter and spring seasons are evaluated. As cited earlier, anomalies were calculated by the difference between seasonal composites and the climatological average of the variables when a FS was over the SRB from 2016 to 2020.

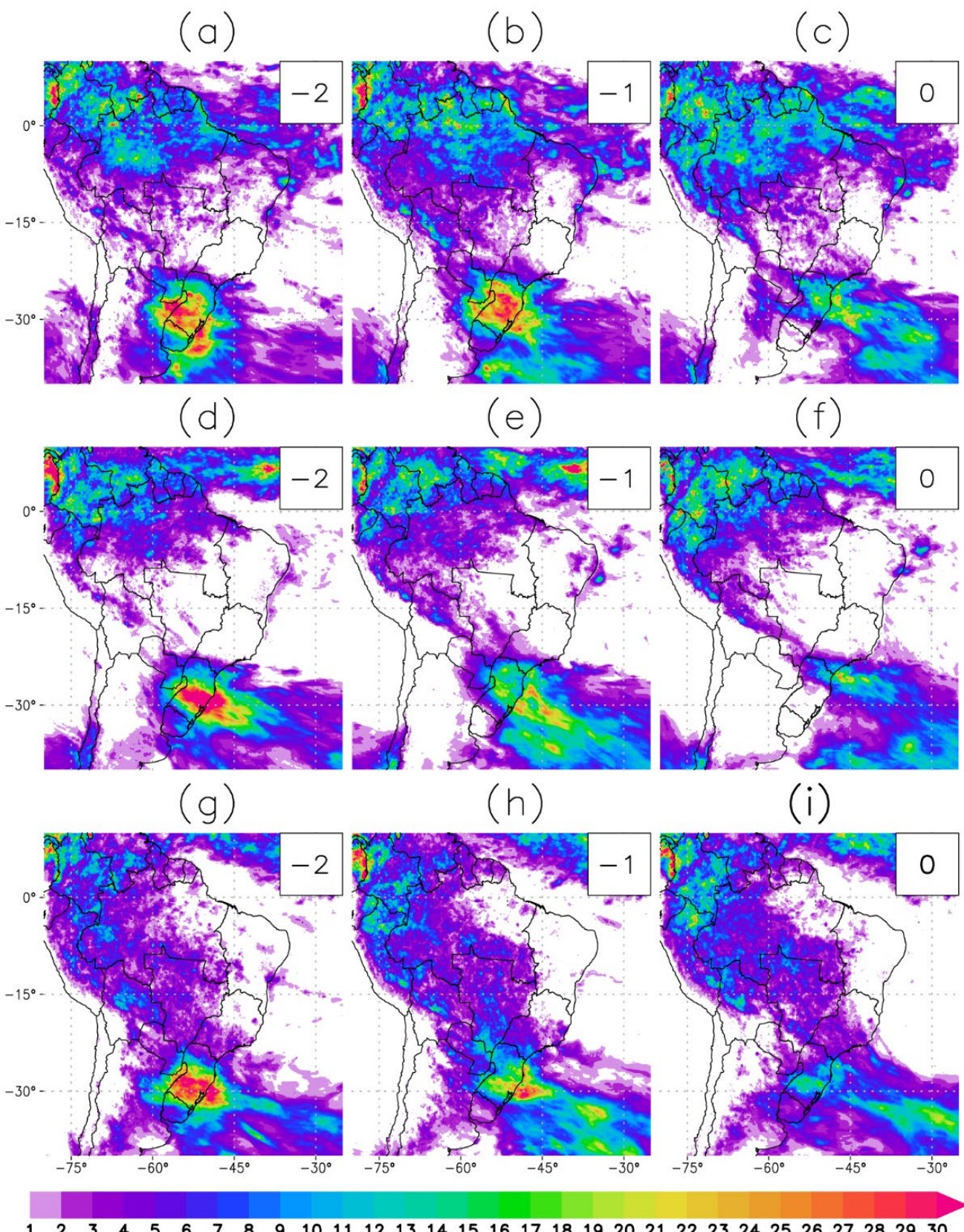

**Figure 5.** Composite precipitation average field (mm day$^{-1}$) in autumn (**a**–**c**), winter (**d**–**f**) and spring (**g**–**i**) at 2 days before (−2), 1 day before (−1) and the day of the event (0).

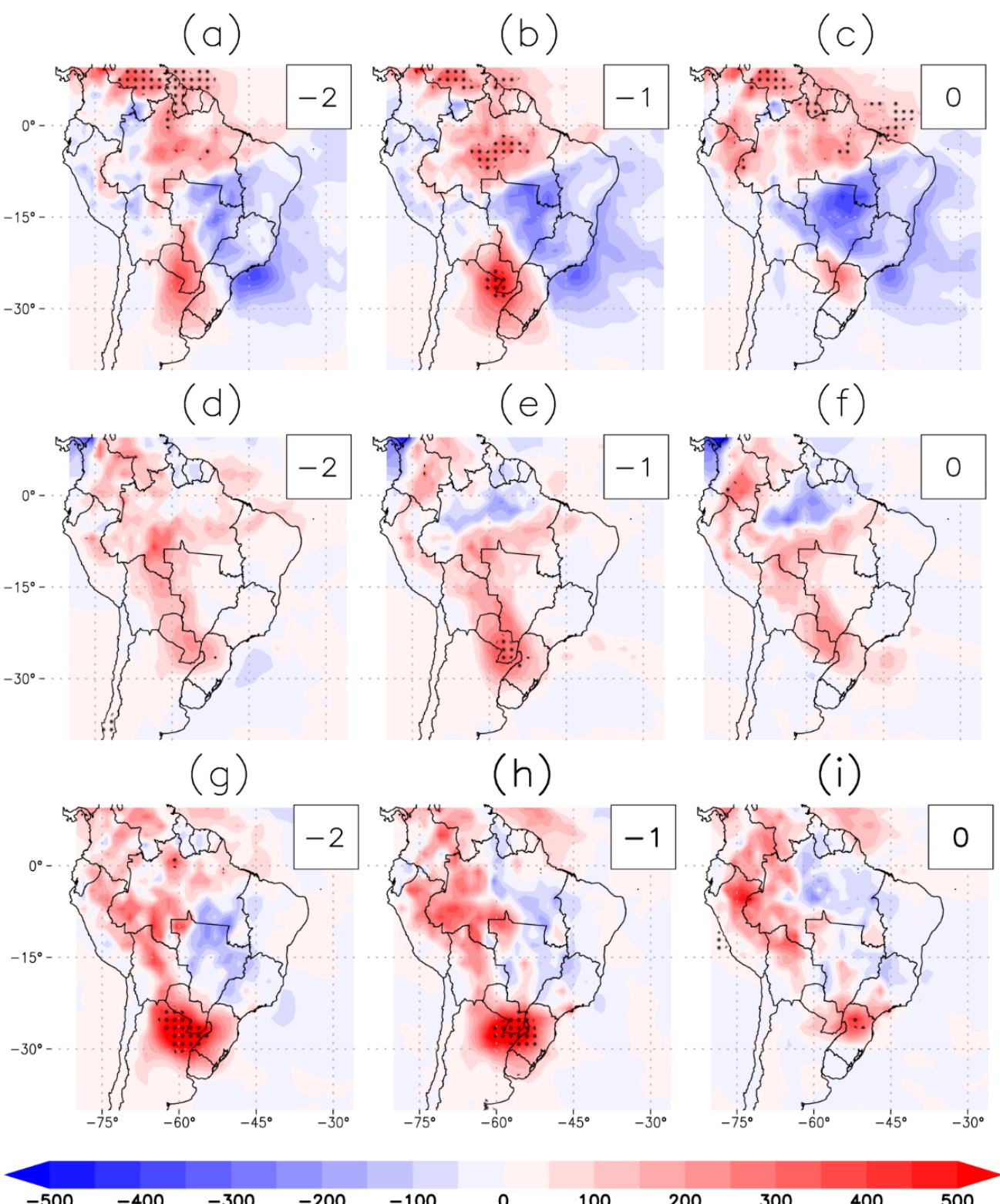

**Figure 6.** Anomaly of the composite of the available convective potential energy average fields (CAPE, in J kg$^{-1}$) in autumn (**a–c**), winter (**d–f**) and spring (**g–i**) at 2 days before (−2), 1 day before (−1) and the day of the event (0). The dotted lines represent the regions with statistical significance at 99%.

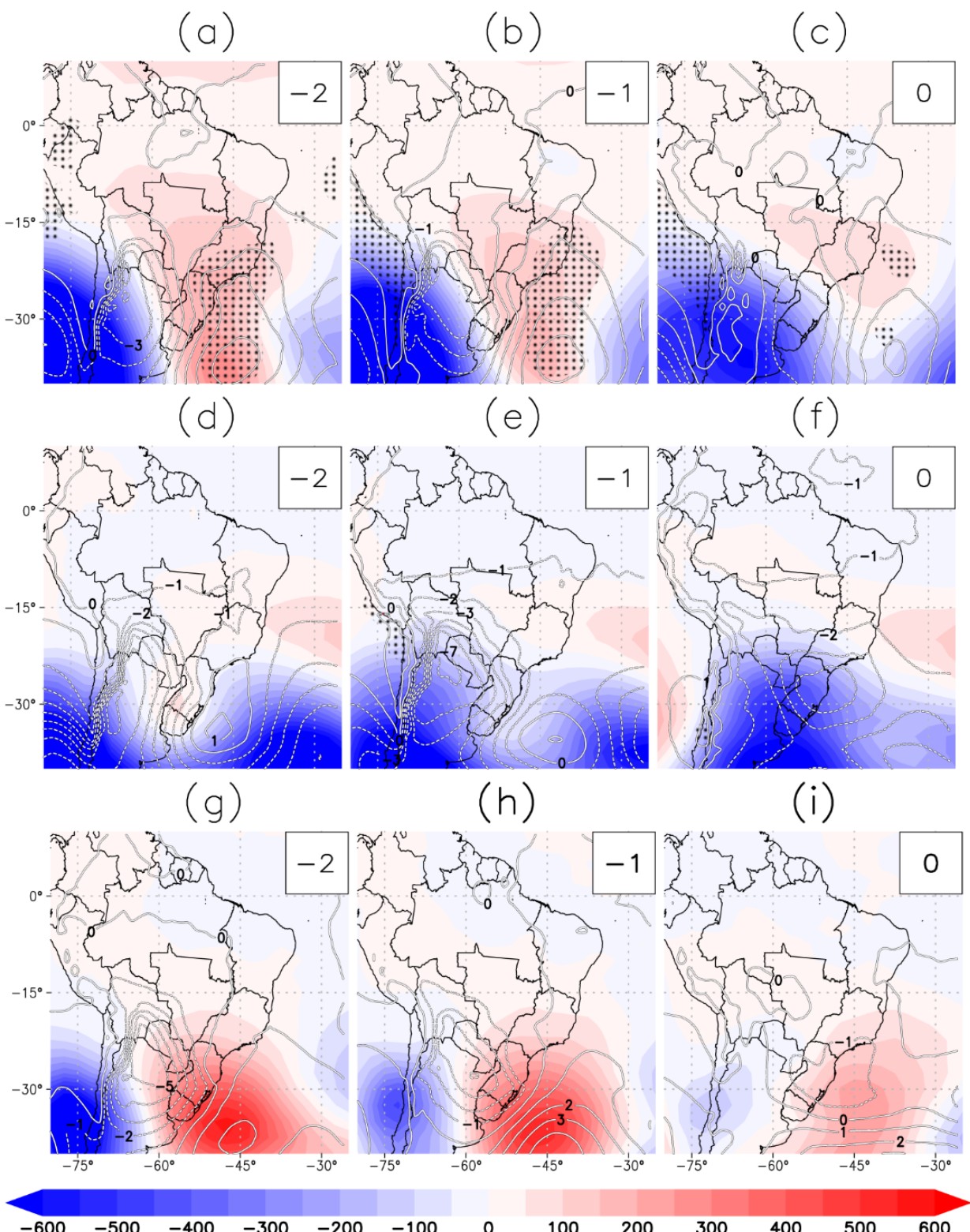

**Figure 7.** The anomaly of the composite of the mean sea level pressure (isolines, hPa) and layer thickness (500–1000 hPa, hatched) in autumn (**a–c**), winter (**d–f**) and spring (**g–i**) at 2 days before (−2), 1 day before (−1) and the day of the event (0). The dotted lines represent the regions with statistical significance at 99%.

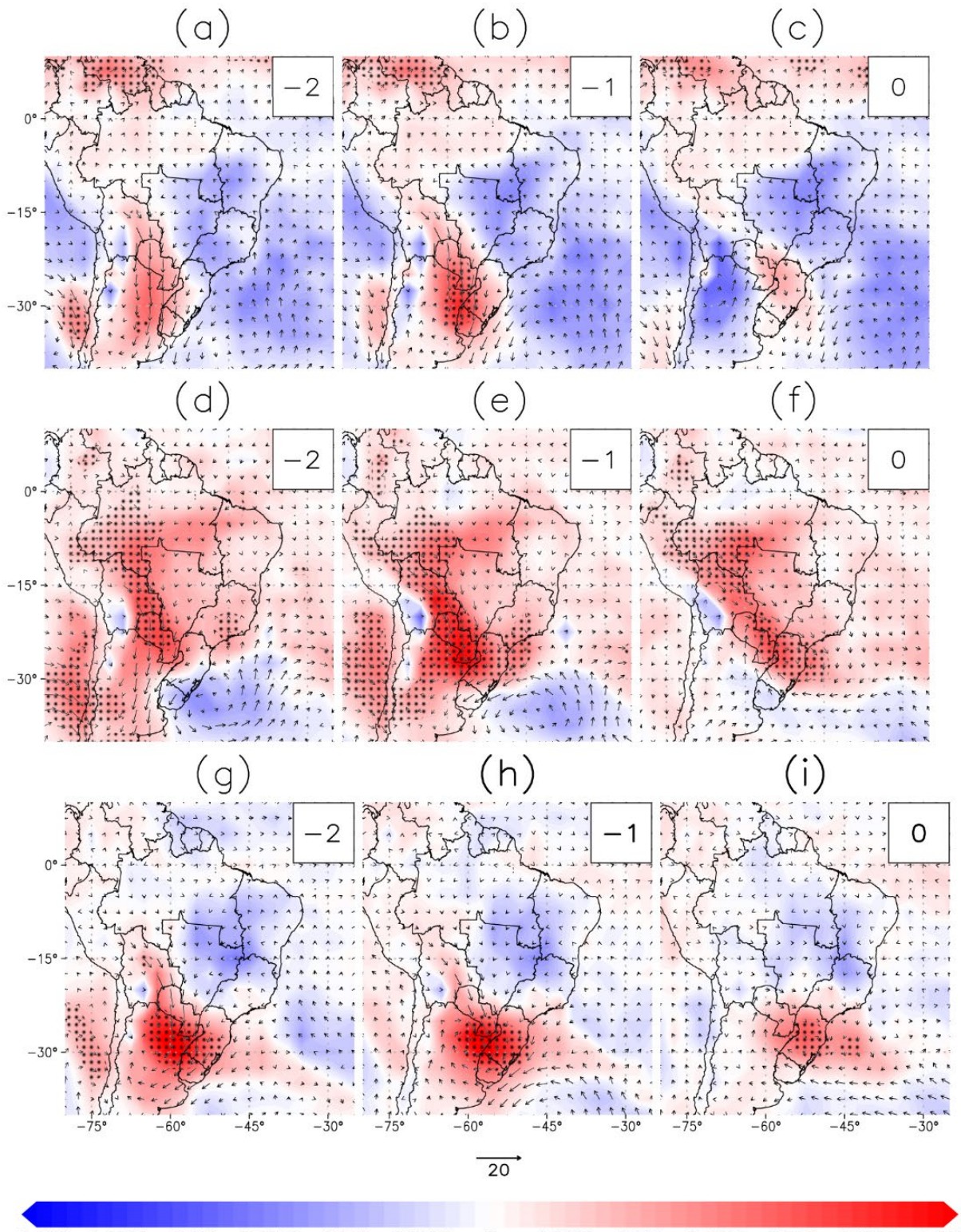

**Figure 8.** The anomaly of the composite of specific humidity average fields (hatched, in kg kg$^{-1}$) and wind direction (arrows) at 925 hPa level in autumn (**a–c**), winter (**d–f**) and spring (**g–i**) at 2 days before (−2), 1 day before (−1) and the day of the event (0). The dotted lines represent the regions with statistical significance at 99%.

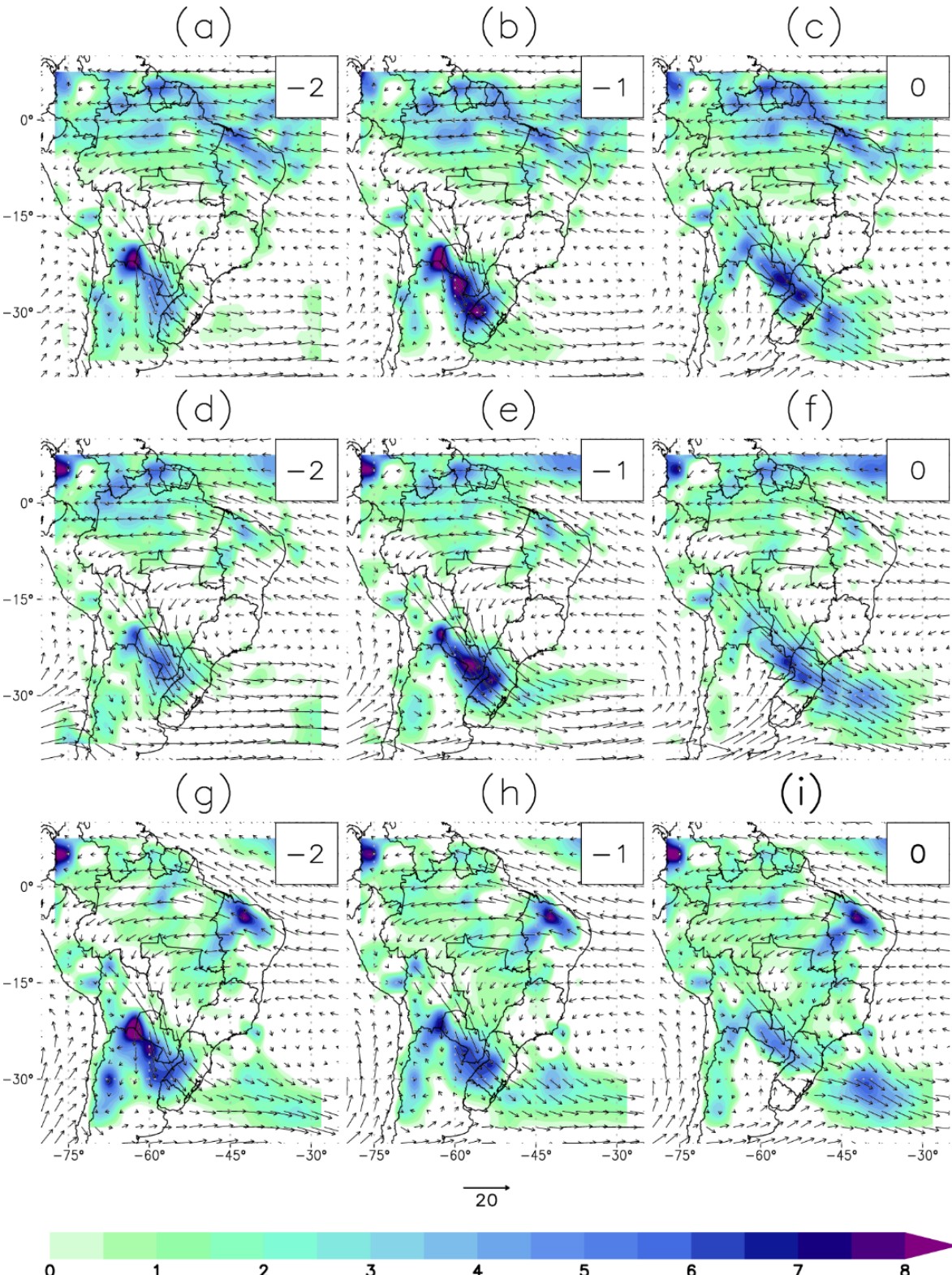

**Figure 9.** Composites of convergence of integrated moisture (kg kg$^{-1}$ s$^{-1}$) in the 1000–700 hPa layer in autumn (**a**–**c**), winter (**d**–**f**) and spring (**g**–**i**).

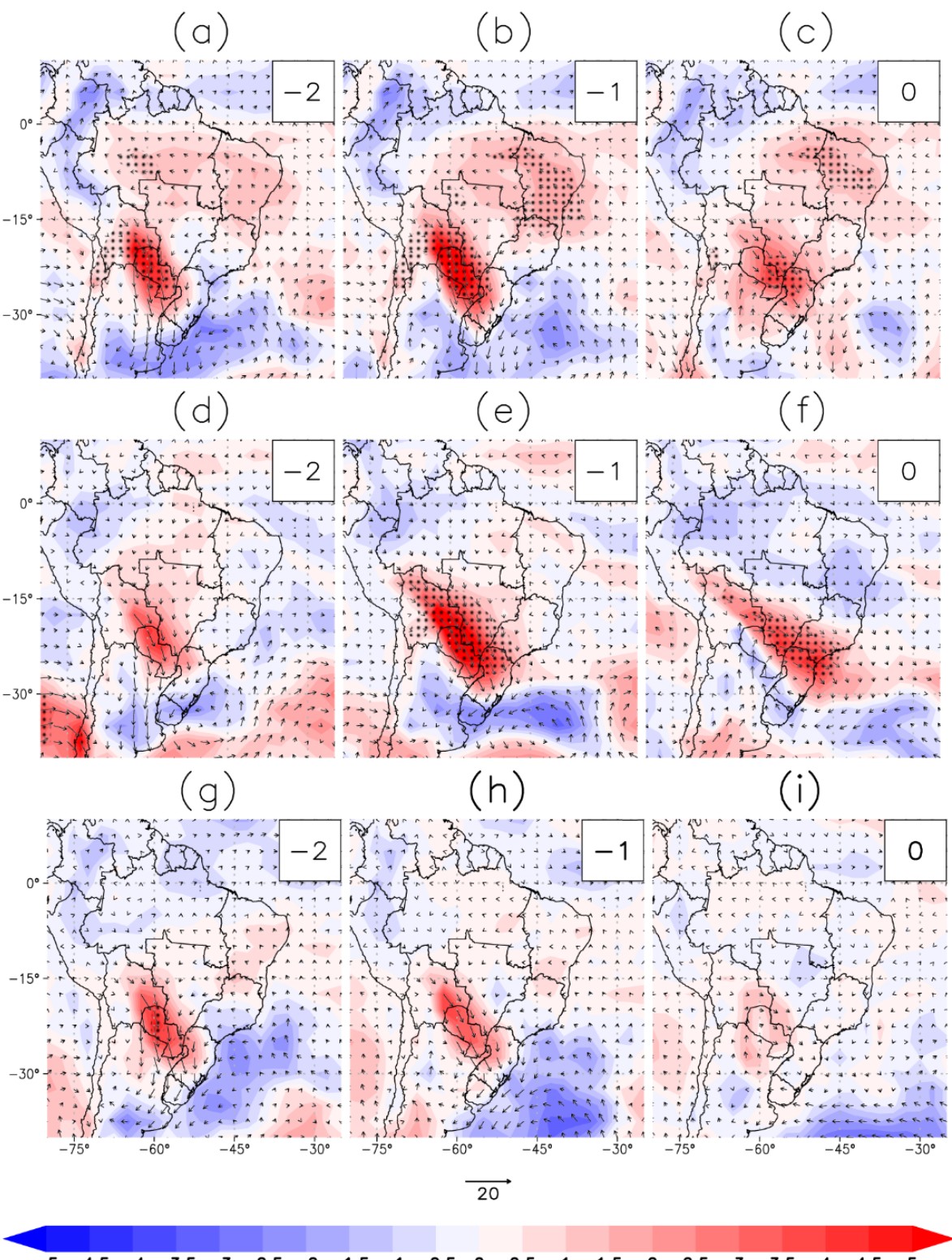

**Figure 10.** The anomaly of the composite of wind speed average fields (hatched in m s$^{-1}$) and wind (arrows) at 850 hPa level in autumn (**a**–**c**), winter (**d**–**f**) and spring (**g**–**i**) at 2 days before (−2), 1 day before (−1) and the day of the event (0). The dotted lines represent the regions with statistical significance at 99%.

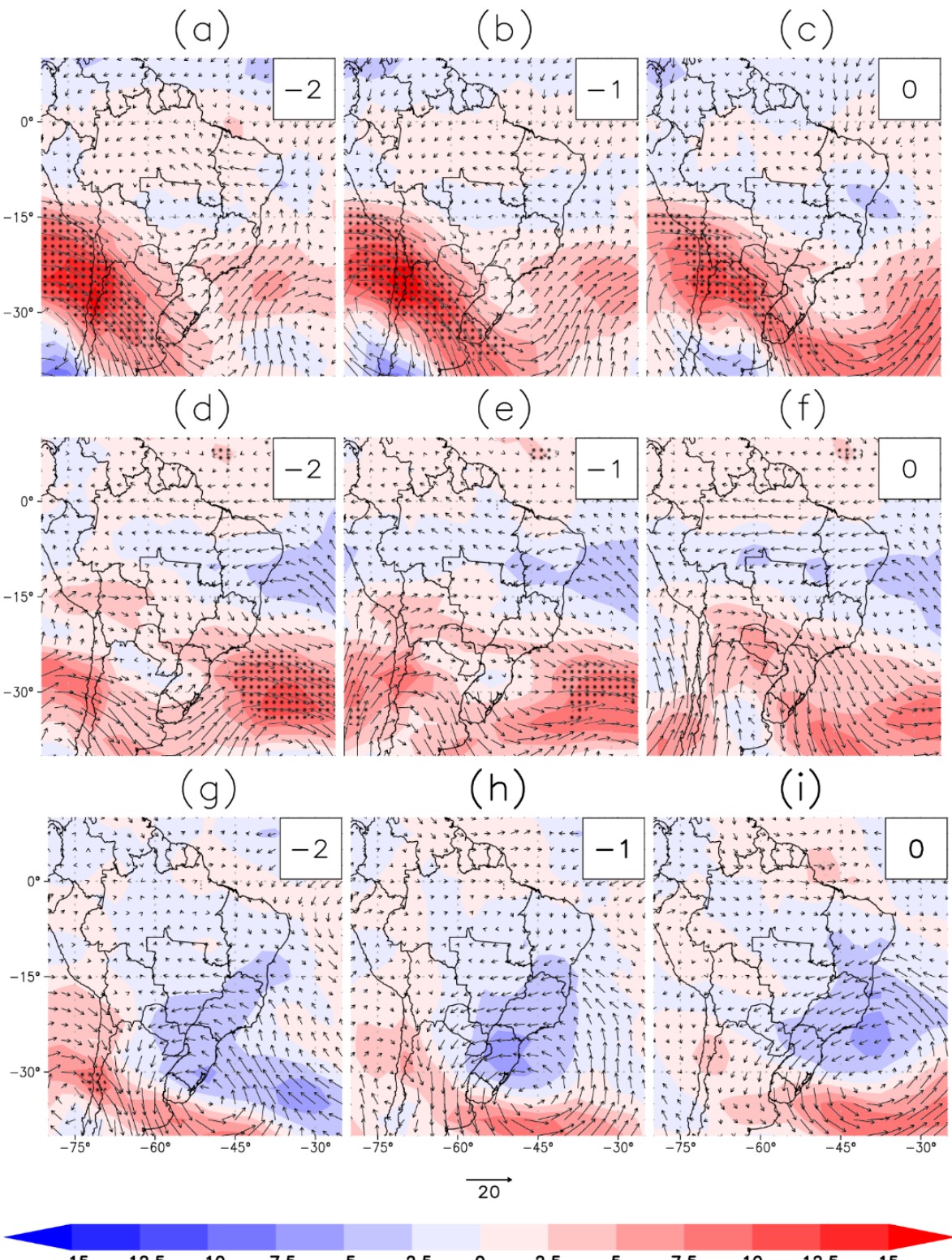

**Figure 11.** The anomaly of the composite of wind speed average fields (hatched in m s$^{-1}$) and wind (arrows) at 250 hPa level in autumn (**a–c**), winter (**d–f**) and spring (**g–i**) at 2 days before (−2), 1 day before (−1) and the day of the event (0). The dotted lines represent the regions with statistical significance at 99%.

The average daily precipitation fields corresponding to the 60 separate days when ND occurred due to FS are shown in Figure 5. Among the seasons, the largest precipitation

accumulations are observed at day −2 and 0 (Figure 5a,d,g). They are concentrated mainly over RS, with emphasis on the northern region and the coast (winter, Figure 5d–f), west (autumn, Figure 5a–c) and center of the State (spring, Figure 5g–i) showing accumulations greater than 30 mm day$^{-1}$. Seluchi [33] analyzed intense rainfall in SC caused by the FS passage and showed precipitation fields similar to those presented in autumn (Figure 5a–c) but with maximum precipitation values of 15 mm day$^{-1}$. While Cavalcanti [35] showed that in spring the highest accumulations were on the day of the FS passage with values around 10 mm day −1. At day −1 (Figure 5b,e,h), precipitation accumulations decrease, except in autumn with values around 30 mm day$^{-1}$ in the states of RS, SC and west of PR. Over the SRB on day 0 (Figure 5c,e,i), the precipitation values are less expressive if compared to the previous days, but the spatial orientation matches the presence of FS, showing values between 20 and 25 mm day$^{-1}$. Significant precipitation accumulations are observed in autumn (Figure 5a–c) and spring (Figure 5g–i) in the eastern region of SC and northern region of RS. This behavior indicates that the rainfall occurring before the FS passage over the SRB causes an accumulation of water in the soil. Thus, even small precipitation accumulations from the FS can initiate an ND. This corroborates previous studies such as Castro [58], which states that flooding occurs when there is accumulation of water in the streets and urban perimeters due to heavy rainfall in cities with deficient drainage systems, and floods occur when there is a gradual increase in water; thus, one disaster is a consequence of another.

To corroborate the analysis of precipitation, the CAPE (J kg$^{-1}$) anomaly fields are shown in Figure 6. The purpose of evaluating the anomaly fields is to determine whether the behavior observed in the composites (−2, −1, and 0—Figure 5) are distinct or similar to the FS climatology. CAPE is used to assess atmospheric instability conditions or convection balance criterion [59]. In general, CAPE values of 1.000 to 2.500 J kg$^{-1}$ are considered high, values above 2500 J kg$^{-1}$ indicate pronounced instability and above 4000 J kg$^{-1}$ indicate extreme instability [60]. In spring, a large area with values of 500 J kg$^{-1}$ above average on day −2 (Figure 7g) and day −1 (Figure 7h) is observed over north-northeastern Argentina, south-central Paraguay and central-western SRB, and the values of 300 J kg$^{-1}$ on day 0 (Figure 6i) over the states of PR and SC. The values and spatial patterns shown agree with the results for the spring composites (Figure 6g–i) and they are statistically significant, confirming that the composites are statistically different when compared to the FS climatology. The spatial pattern of the CAPE anomaly in winter (Figure 6d–f) and autumn (Figure 6a–c), were similar to spring with less pronounced values. Autumn and winter show considerable values as above 400 J kg$^{-1}$ (Figure 6b) and above 200 J kg$^{-1}$ (Figure 6e). In all seasons, one day before ND (−1) was observed in the SRB, the composites are statistically different climatologically. The analysis of the average composite fields of CAPE (J kg$^{-1}$) (Figure S2, Supplementary Materials) in all seasons, a CAPE core is observed between the western SRB, southern Paraguay and northeastern Argentina with values above 1000 J kg$^{-1}$ in spring, 900 J kg$^{-1}$ in autumn and 500 J kg$^{-1}$ in winter. In spring, an extensive area is observed two days before ND with values at 1000 J kg$^{-1}$.

The anomaly of the MPSL fields and layer thickness (hatched) (Figure 7) allow us to clearly observe the displacement of the low pressure (isolines) from west to east at all seasons between 20° and 30° South. In spring (Figure 7g–i), the MSLP anomaly isolines are more defined than in autumn (Figure 7a–c) and winter (Figure 7d–f) due to the greater warming observed in the positive anomaly of the layer thickness. At all seasons, the displacement of a post-frontal high is observed through the layer thickness anomaly fields in the rear of the low pressure (isolines). In autumn, days −2, −1 and 0, we observe regions of statistical significance of the thickness of the layer east of the SRB adjacent to the ocean. It is noticed, mainly in spring, the trough associated with the SF and the pressure gradient associated with the FS and the pressure gradient more to the south of the continent are more intense than the FS climatology, confirmed by the statistical significance of these anomalies over the SRB. Thus, FS that initiate an ND are more intense than the other FS in the region.

A similar behavior of the MSLP average of composite field and the layer thickness 500–1000 hPa is observed during all seasons on all three days (Figure S3, Supplementary Materials). On day −2, a low-pressure center is observed in the Lower Chaco region even in winter when an absence of this system is expected [61]. However, in previous studies, this system is observed 1 day before the arrival of the FS [35]. The significant values of layer thickness in spring in the Chaco region require attention. Spring is a transition season and lower values are expected. Therefore, these results indicate that on days preceding ND, in spring, the atmospheric column is much warmer than FS climatology. On day −1, eastward displacement of the low pressure is observed. Furthermore, it is inferred that the extratropical cyclone associated with the FS on the ND day is located over the South Atlantic Ocean below latitude 40° South and 47° West. By analysis of the thickness 500–1000 hPa field, it is possible to follow the eastward displacement of the FS from day −2 until the ND.

The specific humidity (hatched) and wind (arrows) anomalies at the 925 hPa level are shown in Figure 8. In three seasons (autumn, winter and spring), positive anomalies of specific humidity are observed in the SRB region in the three days. The wind field in winter shows a similar pattern to the jet outflow; anomalously intense winds in the northwest-southeast direction prominently on day −1 (Figure 8e), also observed on day −2 (Figure 8d) and day 0 (Figure 8f). In autumn, a more meridional behavior of the winds is observed on day −2 (Figure 8a) and day −1 (Figure 8b). The spatial behavior of specific humidity anomaly in spring (Figure 11g–i) resembles the spatial behavior of CAPE composites (Figure S2, Supplementary Materials) and CAPE anomaly (Figure 6g–i), an interesting indication of moisture and heat accumulation that enhances the development of convective activity in the SRB. In all autumn and winter days (except autumn −2 and 0), statistical significance is observed where the specific humidity is above the climatological SF over the SRB and over the runoff region of the specific humidity. The anomaly results again confirm that the FS that are causing disasters are different, with the influence of intense runoff over the SRB, causing greater moisture convergence and contribution to convection.

We calculate the convergence of integrated moisture ($kg\,kg^{-1}\,s^{-1}$) in the 1000–700 hPa layer to identify how atmospheric rivers may impact on the formation of heavy precipitation in the region (Figure 9). The results show a similar strong convergence towards the west of the SRB and confluence of the winds in the SRB region on −1 day in autumn and winter. In spring, Figure 9g–i show the highest values of convergence on −2 day. By wind field analysis, convergence of integrated moisture is rather meridional, concentrated in south-central Paraguay on all days analyzed. The wind direction changes, assuming a northwest-southeast direction until day 0.

The analysis of the wind magnitude (hatched) and direction anomaly (arrows) average fields at 850 hPa level (Figure 10) in the autumn, winter and spring show direction and intensity values above SF climatology in the northwest-southeast direction at the SALLJ outflow region mainly one day before ND in all seasons (Figure 10b,e,h). On all days (except day 0—spring) (Figure 10i), a statistical difference between ND events and climatology is observed, including in the region of the SALLJ outflow, especially in winter.

Based on the criteria described by Bonner [62] and adapted for SA by Marengo et al. [63] (wind speed at 850 hPa above $12\,m\,s^{-1}$; vertical shear between 850 and 700 hPa; negative meridional wind component and meridional component modulus greater than the zonal component modulus) on wind speed and wind direction average composite fields at the 850 hPa level (Figure S4, Supplementary Materials), the presence of the South American Low Level Jets (SALLJ) [64–67] was confirmed. SALLJ are observed on days −2 and −1 in autumn and winter with values of wind speed higher than $15\,m\,s^{-1}$. It is not uncommon to observe the occurrence of SALLJ throughout the year. In the warm season, the JBN brings moist tropical air from the Amazon while in the cold season, a season during which the NDs investigated in this study occurred, SALLJ bring tropical maritime air from the subtropical Atlantic Ocean which is less moist than the tropical air masses from the

Amazon [63]. This meridional shift creates conditions for a moist atmosphere northwest of the SRB, which subsequently provides moisture input to a FS which generates an ND.

In all seasons, wind speed anomalies at 250 hPa (Figure 11) are observed: cyclonic circulations in the post-frontal high region discussed previously. In autumn (Figure 11a–c), the arrows indicate northwest-southeast direction in the jet stream region with values 10 m s$^{-1}$ above the FS climatology. In winter (Figure 11d–f), negative anomalous values are observed west of the SRB on day −2 and over the SRB on day −1. In spring (Figure 11g–i), anomalously negative wind speed values are observed on all 3 days, with an easterly to westerly direction. Only autumn showed statistical significance at 99% in the jet region on all days; a confirmation of the composites is statistically larger than the FS climatology in that season. As the anomaly average fields do not show a clear difference between DN derived from FS and common FS, it can be deduced from the observed displacement that the wave front is less intense. Therefore, the system moves more slowly causing precipitation accumulations for longer rather than the FS moving faster.

The average of wind speed composite fields was analyzed for the high levels (250 hPa). Figure S5, Supplementary Materials, shows the presence of the high-level jet [68] which is well established in all three seasons, a well-known feature of SF at high levels of the atmosphere. In winter, as expected, the composition of the cases showed the jet stream positioned between 25° and 35° South over the Atlantic Ocean, with a very zonal behavior and values above 45 m s$^{-1}$, slightly higher than expected [68]. In autumn, when the jet stream starts the intensification process, the expected behavior was observed, being positioned south of the SA and Atlantic Ocean between 30° and 40° South showing wind speed values between 35 and 45 m s$^{-1}$. In spring, the season when the jet stream loses intensity, the jets are positioned lower than the climatological position (30° South). The wind speed values expected in the climatology were observed around 34 m s$^{-1}$, but on the three days of the event composition values above 45 m s$^{-1}$ were identified. In all seasons, a ridge at day −2 positioned between 35° and 40° South and 75° West moving eastward is observed, in agreement with the post-frontal high observed in Figure 7. The clear presence of a high-level anticyclonic circulation similar to the Bolivia High (BA) in all seasons and days, except in spring on day 0 (Figure S5, Supplementary Materials) is detected. In both winter and autumn, this high pressure is located southeast of Amazonas and northwest of Mato Grosso. In winter, the winds at high levels are expected to have zonal characteristics, i.e., no cyclonic or anticyclonic gyres at high troposphere over the BA region, different to the composites in Figure S5 (Supplementary Materials). In spring, the season in which BA begins to appear over the central Amazon, the high is positioned further west of the Amazon.

Descriptive statistical measures were generated for each meteorological variable analyzed (Table 1) and for each county affected by a ND. The precipitation values presented in Table 1 show the average values in spring on day −1 are more intense than day −2 and day 0. While in winter, the highest values are observed on day −2 and low values are observed on day 0. Spring is similar to what was observed in autumn: higher values on day −1, but with lower intensity compared to day −2. In all three seasons the standard deviations were quite similar, ranging between 30 and 45 on days −2 and −1. On the days after ND, little precipitation was recorded (not shown). These results corroborate the spatial pattern observed in Figure 5 and they are strong indications that rains before the arrival of the FS over the region saturate the soil overloading the urban drainage system with a large accumulation of water; thus, the arrival of the FS with the characteristics presented in this study initiate the ND. In addition to the discussion already held on the precipitation values in Table 1, Figure 12 shows the average daily precipitation and allows us to see that in the months of March, May, September and October, the rainfall on the ND day is higher than the climatology. These results show that the rains associated with ND are more intense, generating the potential to cause ND.

**Table 1.** Statistical measures (mean, standard deviation and anomaly) of the meteorological variables studied in each ND affected municipality.

| | Temperature (°C) | | | Specific Hum. (kg kg⁻¹) | | | Pressure (mb) | | | CAPE (J kg⁻¹) | | | Wind Vel (m s⁻¹) | | | Wind Dir (°) | | | Precip. (mm) | | |
|---|---|---|---|---|---|---|---|---|---|---|---|---|---|---|---|---|---|---|---|---|---|
| | Mean | sd | Anom | Mean | sd | Anom | Mean | sd | Anom | Mean | sd | Anom | Mean | sd | Anom | Mean | sd | Anom | Mean | sd | |
| **AUTUMN** | 19.0 | 3.2 | 0.13 | 0.01219 | 0.00218 | 0.00644 | 1016.2 | 3.1 | 0.6 | 190.0 | 329.2 | −3.58 | 2.2 | 1.5 | 1.3 | 140.8 | 120.5 | 65.5 | 31.0 | 41.6 | 2 days before |
| | 19.4 | 3.0 | 0.50 | 0.01280 | 0.00188 | 0.00705 | 1015.5 | 3.6 | −0.1 | 292.1 | 409.0 | 98.52 | 2.4 | 1.2 | 1.5 | 117.8 | 105.0 | 42.6 | 47.9 | 44.1 | 1 day before |
| | 18.6 | 2.9 | −0.22 | 0.01268 | 0.00219 | 0.00693 | 1014.4 | 4.1 | −1.3 | 256.9 | 413.2 | 63.30 | 2.1 | 1.0 | 1.2 | 206.7 | 118.2 | 131.5 | 11.8 | 15.7 | Disaster day |
| | Temperature (°C) | | | Specific hum. (kg kg⁻¹) | | | Pressure (mb) | | | CAPE (J kg⁻¹) | | | Wind vel (m s⁻¹) | | | Wind dir (°) | | | Precip. (mm) | | |
| | Mean | sd | Anom | Mean | sd | anom. | Mean | sd | Anom | Mean | sd | Anom | Mean | sd | Anom | Mean | sd | Anom | Mean | sd | |
| **WINTER** | 14.3 | 2.7 | −4.63 | 0.00906 | 0.00194 | 0.00345 | 1017.4 | 3.7 | 1.8 | 60.7 | 117.5 | −127.0 | 2.4 | 1.8 | 1.7 | 138.7 | 94.2 | 56.3 | 39.9 | 33.5 | 2 days before |
| | 14.5 | 2.8 | −4.38 | 0.00962 | 0.00159 | 0.00401 | 1015.0 | 4.3 | −0.6 | 85.1 | 126.3 | −102.6 | 2.2 | 1.5 | 1.5 | 170.0 | 105.9 | 87.6 | 21.8 | 32.3 | 1 day before |
| | 14.6 | 2.6 | −4.34 | 0.00951 | 0.00180 | 0.00390 | 1013.3 | 4.6 | −2.3 | 77.8 | 80.0 | −109.9 | 2.2 | 1.6 | 1.4 | 198.3 | 94.3 | 115.9 | 2.9 | 7.2 | Disaster day |
| | Temperature (°C) | | | Specific hum. (kg kg⁻¹) | | | Pressure (mb) | | | CAPE (J kg⁻¹) | | | Wind vel (m s⁻¹) | | | Wind dir (°) | | | Precip. (mm) | | |
| | Mean | sd | Anom | Mean | sd | Anom | Mean | sd | Anom | Mean | sd | Anom | Mean | sd | Anom | Mean | sd | Anom | Mean | sd | |
| **SPRING** | 19.2 | 2.6 | 1.23 | 0.01245 | 0.00185 | 0.00689 | 1013.6 | 3.8 | −2.0 | 280.3 | 341.8 | 88.4 | 2.6 | 1.8 | 1.7 | 110.0 | 70.2 | 22.3 | 32.7 | 30.2 | 2 days before |
| | 19.3 | 3.0 | 1.23 | 0.01273 | 0.00197 | 0.00716 | 1013.2 | 4.0 | −2.4 | 342.0 | 397.6 | 150.0 | 2.7 | 2.1 | 1.8 | 119.9 | 83.1 | 32.2 | 38.0 | 45.3 | 1 day before |
| | 18.8 | 2.4 | 0.59 | 0.01229 | 0.00256 | 0.00672 | 1013.7 | 4.5 | −1.8 | 227.8 | 243.1 | 35.9 | 2.8 | 2.0 | 1.9 | 132.7 | 79.6 | 45.0 | 10.7 | 16.9 | Disaster day |

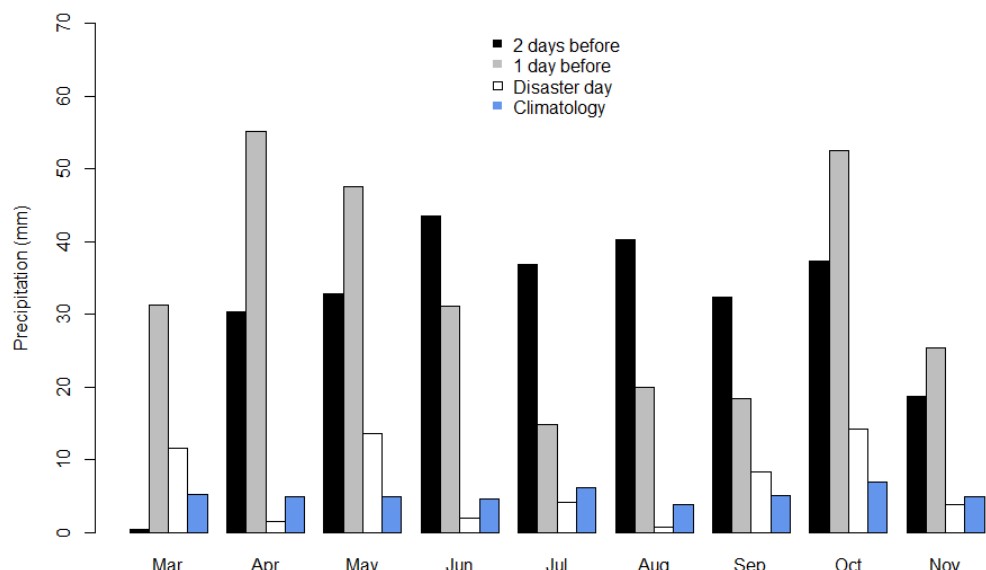

**Figure 12.** Average daily precipitation (mm) in events of Natural Disasters at 2 days before, 1 day before, day of the event and the climatology.

In general, the atmospheric characteristics are consistent with the passage of a FS over the region of interest. Before the arrival of the FS, the temperature increases from day $-2$ to day $-1$ and, consequently, the atmospheric pressure decreases and after the passage of the FS, the temperature continues to decrease while the pressure increases. However, in the spring the minimum atmospheric pressure occurred on day $-1$, contrary to what is expected on the arrival of a FS (Figure 7). Another unusual behavior observed in the spring was the increase in temperature 2 days after the passage of the FS. These results indicate that in the spring, the atmosphere was very warm, corroborated by the anomalous values of temperature, always positive, and atmospheric pressure, always negative, which may be related to the SALLJ and the consequent moisture input in the region. Another unusual result is the sharp temperature anomaly values in winter, 4 °C lower than the climatological average of the FS, indicating how cold the atmosphere was before and during the ND.

During all seasons, the wind direction was in the southeast quadrant (90° to 180°) with wind speed values around 2.4 m s$^{-1}$ and values 1.5 m s$^{-1}$ above average. The specific humidity (kg kg$^{-1}$), at all stations, increased from day $-2$ to day $-1$ and decreased from day $-1$ to day 0, this behavior is related to the more intense rains occurring on day $-1$, decreasing the humidity present in the atmosphere. On all days and in all seasons the specific humidity values were above the climatological average, around 0.00584 kg kg$^{-1}$.

The average CAPE values corroborate the spatial behavior observed in Figures 6 and 7. Spring shows very pronounced mean values, indicating an atmosphere with pronounced instability and more intense than the other seasons. For most of the variables analyzed, the standard deviations were low, except CAPE and wind direction due to variables' variability. CAPE can change rapidly between convection formation (high CAPE values) and precipitation (low CAPE values) while wind direction, being measured in degrees, alternates the values of the north direction between 0° and 360°, which strongly influences the averaging.

## 4. Conclusions

In this work, an analysis of weather conditions associated with FS that triggered ND over the SRB from 2016 to 2020 was performed. ND occur mostly in winter (June and July) and spring (October), which is consistent with the literature [33,35,49]. It was found that the FS that trigger an ND are statistically different to other FS that pass over the SRB. From the analysis of the composites combined with the anomalies of meteorological variables, it was found that the pressure gradient associated with the FS that initiates an ND is more

intense and the presence of the SALLJ generates moisture convergence and intensifies convection in the region. The high level analysis showed the frontal trough moves more slowly causing the FS to influence the weather conditions for a longer time. The analysis of the composites and anomalies was crucial to determine certain spatial patterns that exist in the atmosphere on the days preceding and on the day that the ND occurs, such as:

- High CAPE values in the western SRB, especially in spring;
- Anticyclonic circulation in the high troposphere, similar to BA in all seasons;
- An atmosphere that is warmer in spring and colder in winter;
- Wind flow at 850 hPa due to SALLJ carrying specific moisture from low latitudes and jet streams with slightly higher wind speed values than climatology.

Considering that CAPE is increased by low level warm air advection and low-level moisture advection (high low-level dewpoint), the results show that from day −2, the atmospheric rivers induced by SALLJ generate humidity convergence in low levels and encounter an anomalous warming in west of the SBR. Thus, from day −2, the FS advances over the region with a favorable atmosphere to convection and precipitation.

With regards to precipitation behavior, in autumn, precipitation averages are highest on day −1 (47.9 mm). In spring, precipitation averages on the days preceding the ND are around 30~35 mm. In winter, average rainfall on days −2 and −1 is observed around 39 and 21 mm, respectively. On day 0, the average precipitation values are 2.9 mm in winter, 11.8 mm in fall, and 10.7 in spring. These results show a strong indication about the municipalities. These locations already witnessed rainfall that saturated the soil before the arrival of the precipitating system that triggered the disaster. Thus, the combination of the mean values and atmospheric behavior presented in this study creates great potential for ND if an FS passes through the region of interest. In addition, the precipitation values presented were higher than the literature values in ND over the SRB [33,35].

The intensity of the precipitation is evident when comparing the daily averages for the days preceding the ND and the climatology (Figure 12). In terms of comparison, the values of precipitation on the day before disasters days are ten times greater than the climatology. Only on disasters days, the daily average precipitation values are less than climatology.

The seasons were analyzed individually, and each showed unique patterns; however, autumn and spring showed similar patterns to winter. In autumn, daily accumulated precipitation values above 30 mm are observed on days −2 and −1 before the passage of the FS concentrated over much of RS and SC. On the day of the ND, rainfall is concentrated on the coast of SC with values between 20 and 25 $mm^{-1}$. In the three days, an intense convergence is observed at low levels with values around 0.013 kg $kg^{-1}$ with meridional winds in the two days preceding the arrival of the FS and later change to a northwest-southeast direction. The jet stream at high levels is positioned further south of the SRB and the presence of SALLJ days before the ND with wind speeds between 13 and 14 m $s^{-1}$.

However, other characteristics such as anticyclonic circulation at high levels and a low-pressure system similar to the Baja del Chaco associated with a convergence of humidity caused by SALLJ that transports moisture from the tropics to the SRB are particularities seen in the days preceding ND in the SRB and can help preventive and mitigation actions of these events in this season.

In spring, warmer days are associated with ND in the SRB as observed by the temperature and layer thickness fields. Although SALLJ is not configured according to the literature, there is still runoff transporting moisture from low latitudes to the SRB even in smaller magnitude. At high levels, the jet stream that should be weakened is active with values above 45 m $s^{-1}$. CAPE is the variable that requires more attention in the spring season, because of the significant values observed (above 1000 J $kg^{-1}$) and the extensive area affected that covers the north and northeast of Argentina, south of Paraguay and west of the SRB. This pattern of anomalously high temperatures, convergence of humidity in the region and high values of CAPE can trigger the rains that begin days before the passage of the FS that causes ND.

**Supplementary Materials:** The following supporting information can be downloaded at: https: //www.mdpi.com/article/10.3390/atmos13111886/s1, Figure S1: Surface synoptic charts (top panel) and satellite image (bottom panel) of 13 October 2017, one of the NDs analyzed in this work, Figure S2: Composites of the available convective potential energy average fields, Figure S3: Composites of the mean sea level pressure (isolines, hPa) and layer thickness average fields (500–1000 hPa, hatched), Figure S4: Composites of wind speed average fields (hatched in m s$^{-1}$) and wind (arrows) at 850 hPa level, Figure S5: Similar to Figure S4, but at 250 hPa level, Figure S6: Surface synoptic charts (top panel) and satellite image (bottom panel) of 13 October 2017, one of the NDs analyzed in this work

**Author Contributions:** Conceptualization, J.S.d.R., W.A.G. and D.M.; methodology, J.S.d.R. and W.A.G.; software, J.S.d.R.; data curation, J.S.d.R. and D.O.d.S.; writing—original draft preparation, J.S.d.R.; writing—review and editing, J.S.d.R., W.A.G., D.M. and D.O.d.S. All authors have read and agreed to the published version of the manuscript.

**Funding:** This research received no external funding.

**Institutional Review Board Statement:** Not applicable.

**Informed Consent Statement:** Not applicable.

**Data Availability Statement:** The data used in this manuscript are available by writing to the corresponding authors.

**Acknowledgments:** The authors thank the National Center for Monitoring and Alerts for Natural Disasters—CEMADEN for the data provided. We are thankful to the National Council for Scientific and Technological Development (CNPq) and the Coordenação de Aperfeiçoamento de Pessoal de Nível Superior—Brasil (CAPES) for the research grant of the first author.

**Conflicts of Interest:** The authors declare no conflict of interest.

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
