# Peer review of "Evaluation of Atmospheric Features in Natural Disasters due Frontal Systems over Southern Brazil"

_atmosphere, doi:10.3390/atmos13111886_

Round 1

Reviewer 1 Report

1. English language and the use of established expressions and terminology.

2. Since the article considers only cases associated with heavy precipitation, it is not clear how heavy or intense these precipitations must be in order to cause an ND. Table 1 gives information on precipitation, but it is not clear how they extreme is compared to climate averages (e.g.,  % of climate normal).

3. The terminology

- MCS  Mesoscale Convective Systems | NASA Airborne Science Program – not CMS (row 218).

-           layer thickness between 1000 and 500 hPa??  It is commonly referred as “thickness 500-1000 hPa” or “500-1000 hPa relative topography”

4. The article talks about the influence of typical synoptic processes and fronts on weather conditions. Was the position of the fronts relative to study area typical to recommend this as a prognostic pattern?

Possibly, it is necessary to objectively determine the average position of atmospheric fronts as zones of high temperature gradients or using, for example, a thermal frontal parameter (TFP).

5. It is desirable to give typical examples of synoptic situations illustrated by weather maps  and satellite information, confirming the formation of the MCS's in the fronts areas.

6. Figures 14-17 duplicate each other, because the zones of convergence and divergence are already detect in the wind fields.

7. Humidity and wind fields show that atmospheric rivers may impact on the formation of heavy precipitation in the region. The calculation of the integral water vapor  transport (IVT) could qualitatively improve the presented study and replace the analysis of individual fields of wind and specific humidity.

8. Fig. 18 - the time course of temperature and other characteristics in the frontal zone is determined by the type of front, and therefore is of no interest, since it is already known.

9. In the conclusions. From the study, it is not clear what mechanism leads to the realization of CAPE, because the convective precipitation, according to the last paragraph of the conclusions, begins before the passage of the front. It is necessary to more accurately find the relationship of all components.

Reviewer 2 Report

The article contains a lot of figures, it is enough to give figures with the Anomaly of the composite of average fields, so figures 6, 8, 10, 12, 14, 16 do not need to be presented.

Reviewer 3 Report

This paper analyzed the weather conditions associated with the frontal systems which triggered natural disasters over SRB during a certain period. Several datasets are used to implement the analysis. The main concerns about this manuscript are summarized as following:

1.     The current version suffers from lack of novelty and contribution. The methods or models used in this paper are basic and commonly adopted in the field of statistics, no proposed methods are found in the manuscript.

2.     The contribution of the analysis results is limited, the conclusions drawn are somehow superficial and general. More specific data analysis is needed to support the conclusion.

3.     The logo of an organization appeared in Fig. 2, which looks a little weird.

4.     There are obvious editing errors in the reference section.

Round 2

Reviewer 1 Report

Editing of English language and style still required.

Reviewer 3 Report

My queries are addressed successfully. I recommend this work for possible publication.